# MQuAKE: Assessing Knowledge Editing in Language Models via Multi-Hop Questions

**Zexuan Zhong**[†*], **Zhengxuan Wu**[‡*],
**Christopher D. Manning**[‡], **Christopher Potts**[‡] and **Danqi Chen**[†]
[†]Princeton University    [‡]Stanford University
{zzhong,danqic}@cs.princeton.edu  {wuzhengx,manning,cgpotts}@stanford.edu

## Abstract

The information stored in large language models (LLMs) falls out of date quickly, and retraining from scratch is often not an option. This has recently given rise to a range of techniques for injecting new facts through updating model weights. Current evaluation paradigms are extremely limited, mainly validating the recall of edited facts, but changing one fact should cause rippling changes to the model's related beliefs. If we edit the UK Prime Minister to now be *Rishi Sunak*, then we should get a different answer to *Who is married to the British Prime Minister?* In this work, we present a benchmark, MQuAKE (**M**ulti-hop **Qu**estion **A**nswering for **K**nowledge **E**diting), comprising multi-hop questions that assess whether edited models correctly answer questions where the answer should change as an entailed consequence of edited facts. While we find that current knowledge-editing approaches can recall edited facts accurately, they fail catastrophically on the constructed multi-hop questions. We thus propose a simple memory-based approach, MeLLo, which stores all edited facts externally while prompting the language model iteratively to generate answers that are consistent with the edited facts. While MQuAKE remains challenging, we show that MeLLo scales well with LLMs (up to 175B) and outperforms previous model editors by a large margin.[1]

## 1 Introduction

As large language models (LLMs) are deployed widely, the need to keep their knowledge correct and up-to-date without massive retraining costs becomes increasingly important (Sinitsin et al., 2020). Prior work has proposed knowledge editing methods to incrementally inject a set of new facts into a language model (Zhu et al., 2021; De Cao et al., 2021; Meng et al., 2022a,b; Mitchell et al.,

---

[*]Equal contribution.
[1]Our datasets and code are publicly available at https://github.com/princeton-nlp/MQuAKE.

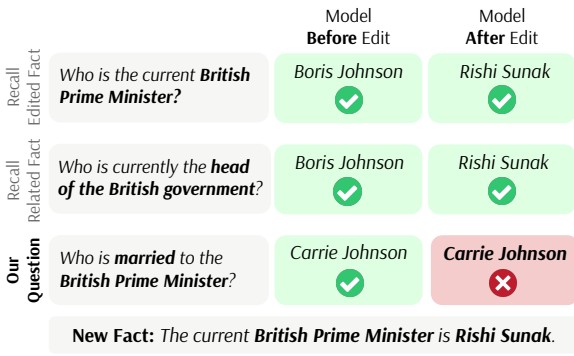

Figure 1: An example of our benchmark MQuAKE. Existing knowledge-editing methods often perform well at answering paraphrased questions of the edited fact but fail on multi-hop questions that are entailed consequences of the edited fact.

2022a,b), but it is not yet clear whether these methods provide a viable solution of updating and maintaining deployed LLMs.

To evaluate these methods, existing benchmarks often focus on measuring whether the edited model can recall the newly injected facts and whether unrelated knowledge remains unchanged post-editing. However, a vital unaddressed question is whether the edited model can handle questions where the answer should change as an entailed consequence of edited facts. For example (see Figure 1), if we update the British Prime Minister from *Boris Johnson* to *Rishi Sunak* within a model, the answer to *Who is married to the British Prime Minister?* should also change as a consequence of this edit.

Therefore, we propose MQuAKE (**M**ulti-hop **Qu**estion **A**nswering for **K**nowledge **E**diting), a new benchmark for a more complete evaluation of knowledge-editing methods. Each example in MQuAKE consists of a multi-hop question (including {2, 3, 4}-hop questions) which corresponds to a chain of facts. When we edit one or a few facts in a chain, the edited model needs to propagate the change to entailed consequences of the edited facts. MQuAKE includes a dataset MQuAKE-CF

based on counterfactual edits, and another dataset MQUAKE-T of temporal knowledge updates to evaluate model editors on real-world changes.

We evaluate state-of-the-art knowledge-editing methods on MQUAKE, testing from editing facts mentioned in one question to editing facts mentioned in a large set of questions. The latter setting evaluates approaches that are designed to handle many edits, such as MEMIT (Meng et al., 2022b). Surprisingly, existing knowledge-editing methods often perform well on answering questions that are paraphrases of the edited fact but fail drastically on questions where the answer should change as a consequence of an edited fact. For example, a GPT-J model edited by ROME (Meng et al., 2022a) can only answer $7.6\%$ of multi-hop questions in MQUAKE-CF, even though it could answer $43.4\%$ of the questions before editing.

Towards faithful knowledge editing, we propose a simple but effective method, MeLLo, that significantly outperforms existing model editors even with a large number of edits. Instead of updating model weights, MeLLo stores edits in an explicit memory inspired by memory-based editing methods (Mitchell et al., 2022b) and prompts the language model iteratively to interact with the edited facts. Specifically, it decomposes a multi-hop question into sub-questions successively, generates tentative answers, and checks whether it is consistent with edited facts before returning the final answer. Such a method does not require any additional training, and can be easily scaled up to large LMs such as GPT-3 (Brown et al., 2020; Ouyang et al., 2022) unlike methods requiring weight updates. We hope that both our benchmark and proposed method provide additional insights into building faithful knowledge-editing methods.

## 2  Problem Definition

This section introduces our setting and argues that a model edit is only truly successful if the edited model also returns new correct answers for questions that change as a consequence of the edits.

### 2.1  Querying Factual Knowledge in LLMs

We represent a fact as a triple $(s, r, o)$, consisting of a subject $(s)$, a relation $(r)$, and an object $(o)$, and manually construct a natural language prompt template $t_r(\cdot)$ for each type of relation $r$ as a way of querying knowledge from a language model (Petroni et al., 2019). This template

takes a subject $s$ as input and generates a question or a cloze-style statement $t_r(s)$. For instance, given the subject *United Kingdom* and the relation *head of government*, we can form a cloze sentence *The Prime Minister of the United Kingdom is __*.

We consider an autoregressive language model $f: \mathcal{X} \to \mathcal{Y}$, which takes a piece of text $x \in \mathcal{X}$ as input and predicts $y \in \mathcal{Y}$, the continuation of $x$. Given a fact triple $(s, r, o)$, we can query the language model to recall the fact by feeding the prompt $t_r(s)$ as input and matching the output $f(t_r(s))$ with the object $o$. While prior work has studied how to prompt to extract more knowledge (Jiang et al., 2020; Shin et al., 2020; Zhong et al., 2021), we simply use manually-written templates, as enhancing knowledge retrieval is not our focus.

### 2.2  Knowledge Editing

The knowledge stored in a language model can be incorrect or become outdated over time. One possible solution is to edit the knowledge on the fly without retraining. A fact edit is defined as a pair of fact triples that share the same subject and relation $e = ((s, r, o), (s, r, o^*))$, which represents the associated object is updated from $o$ to $o^*$. For simplicity, we abbreviate the notation for an edit as $e = (s, r, o \to o^*)$ throughout the paper.

Given a collection of fact edits $\mathcal{E} = \{e_1, e_2, \dots\}$ and a language model $f$, knowledge editing involves learning a function $K : \mathcal{F} \times \mathcal{E} \to \mathcal{F}$ that yields an edited language model $f^* : \mathcal{X} \to \mathcal{Y}$, $K(f, \mathcal{E}) = f^*$. For the methods we assess in Section 4, $K$ modifies the weights of $f$ in an attempt to incorporate $\mathcal{E}$. Our proposed alternative, MeLLo, is much more lightweight: it keeps $f$ frozen and instead uses $\mathcal{E}$ as an external knowledge store to guide generation (Section 5).

In previous work (De Cao et al., 2021; Mitchell et al., 2022a,c; Meng et al., 2022a,b), the evaluation focuses on assessing whether the edited model recalls the updated knowledge and whether unrelated knowledge remains unchanged post-editing. To evaluate whether a "single-hop" edit $e = (s, r, o \to o^*)$ is successful with an edited model $f^*(\cdot)$, existing paradigms assess whether $f^*(t_r(s))$ is equal to $o^*$ (or assigns $o^*$ a high probability). Additionally, they check correctness by varying $t_r(s)$ while keeping semantic equivalence (Meng et al., 2022b).

### 2.3  Evaluation of Multi-hop Questions

By only evaluating single-hop questions, existing methods were tested for recalling edited facts.

It remains unknown whether an edited model can handle a question where the answer should change as an entailed consequence of an edited fact. We propose to evaluate edited models with multi-hop questions by considering a *chain* of facts $\mathcal{C} = \langle (s_1, r_1, o_1), \ldots, (s_n, r_n, o_n) \rangle$, where the object of $i^{\text{th}}$ fact also serves as the subject of the next fact in the chain, i.e., $o_i = s_{i+1}$. We denote $\mathcal{R} = [r_1, \ldots, r_n]$ as the relation set and $\mathcal{S} = [s_1, \ldots, s_n]$ as the subject set. We then use $\mathcal{C}$ to construct a multi-hop question that asks about the head entity $s_1$, with the answer being the tail entity $o_n$. Similar to a single-hop question, we generate a question as $t_{\mathcal{R}}(\mathcal{S})$. For example, with a chain consisting of two facts (*United Kingdom*, *head of government*, *Boris Johnson*), (*Boris Johnson*, *spouse*, *Carrie Johnson*), one can write a 2-hop question *Who is married to the British Prime Minister?* Once one or more facts in the chain are edited, e.g., (*United Kingdom*, *head of government*, *Boris Johnson → Rishi Sunak*), the language model has to leverage the updated knowledge to answer the question, which we posit as a crucial indicator of a model faithfully updating the fact.

## 3 MQUAKE: Multi-hop Question Answering for Knowledge Editing

We construct the benchmark MQUAKE (**M**ulti-hop **Qu**estion **A**nswering for **K**nowledge **E**diting), which contains two datasets. The first, MQUAKE-CF, is designed as a diagnostic dataset for the evaluation of knowledge editing methods on counterfactual edits. The second, MQUAKE-T, comprises temporal-based knowledge updates and is intended to assess the effectiveness of knowledge editing methods in updating outdated information with current, real facts. We first present the data construction process for MQUAKE-CF and MQUAKE-T. Then, we present the data statistics and evaluation settings, followed by evaluation metrics in the end.

### 3.1 Data Construction of MQUAKE-CF

**Sampling chains of facts** Our dataset is constructed based on Wikidata (Vrandečić and Krötzsch, 2014), a knowledge base consisting of fact triples associated with millions of entities. We first sample chains of facts from Wikidata. We manually select 37 common relations and consider a subgraph that solely comprises these relations and the top 20% of common entities based on hyperlink counts in Wikipedia articles.[2] We collect chains that contain $N = 2, 3, 4$ triples from the Wikidata subgraph. We also adopt heuristics rules to ensure that the sampled fact triples are coherent and lead to natural questions (see Appendix A.1 for details).

**Filtering unrecallable facts** As answering multi-hop questions requires the model to leverage each single-hop fact, we filter out any chain of facts which contain at least one fact that cannot be recalled by GPT-J (Wang and Komatsuzaki, 2021), which we will mainly evaluate on. To recall single-hop facts, we curate a question template for each relation type following prior work (Petroni et al., 2019; Meng et al., 2022a), and query the model using in-context learning with 8 demonstration examples (see Appendix A.2 for more details).

**Generating multi-hop questions** Given $\mathcal{C} = \langle (s_1, r_1, o_1), \ldots, (s_n, r_n, o_n) \rangle$, we aim to write a set of questions $\mathcal{Q}$ about the head entity $s_1$ with the gold answer $a$ being the tail entity $o_N$. We leverage ChatGPT (gpt-3.5-turbo) to automatically generate questions given a chain of facts $\mathcal{C}$, because (1) this provides us more diverse question formats of good quality; (2) it is challenging to manually write question templates for all the different types. We prompt ChatGPT to generate *three* questions for each sampled chain of facts. We include the prompt we used and examples of generated multi-hop questions in Appendix A.3.

**Sampling counterfactual edits** So far, we have collected $\langle \mathcal{Q}, a, \mathcal{C} \rangle$ (questions, answer, fact triples) for each instance in the dataset. Next, we sample counterfactual edits $\mathcal{E}$ and collect the corresponding fact triples $\mathcal{C}^*$ and answer $a^*$. Given a chain of $n$ factual triples $\mathcal{C} = \langle (s_1, r_1, o_1), \ldots, (s_n, r_n, o_n) \rangle$, we randomly sample $t \in \{1, \ldots, N\}$ counterfactual edits in $\mathcal{C}$. For a triple $(s, r, o)$, we sample a counterfactual object $o^*$ from all possible objects that are related to relation $r$. We replace $(s, r, o)$ with $(s, r, o^*)$ in the chain and update other facts accordingly. We make sure that, after injecting counterfactual edits, the new chain still exists so that we are able to find an updated answer $a^*$. We only keep the sampled edits if the corresponding updated answer $a^*$ is not identical to the original one $a$. We use the same filtering process as Appendix A.2 to make

---

[2]We focus on common entities as LMs can recall facts about common entities more reliably (Mallen et al., 2023).

| | | | | | |
|---|---|---|---|---|---|
| $\mathcal{E}$ | (WALL-E, creator, Andrew Stanton → James Watt) | | | | |
| | (University of Glasgow, headquarters location, Glasgow → Beijing) | | | | |
| $\mathcal{Q}$ | In which city is the headquarters of the employer of WALL-E's creator located? | | | | |
| | What is the location of the headquarters of the company that employed the creator of WALL-E? | | | | |
| | Where is the headquarters of the company that employed the creator of WALL-E situated? | | | | |
| $a$ | Emeryville | | | | |
| $a^*$ | Beijing | | | | |
| $\mathcal{C}$ | (WALL-E, creator, Andrew Stanton) | | | | |
| | (Andrew Stanton, employer, Pixar) | | | | |
| | (Pixar, headquarters location, Emeryville) | | | | |
| $\mathcal{C}^*$ | (WALL-E, creator, James Watt) | | | | |
| | (James Watt, employer, University of Glasgow) | | | | |
| | (University of Glasgow, headquarters location, Beijing) | | | | |

Table 1: An instance in the MQUAKE-CF dataset, which consists of an edit set $\mathcal{E}$, a set of three multi-hop questions $\mathcal{Q}$, the desirable answer pre- and post-editing $a, a^*$, and the chain of facts pre- and post-editing $\mathcal{C}, \mathcal{C}^*$. The edited facts are marked as $(\underline{s}, \underline{r}, \underline{o}^*)$.

| | #Edits | 2-hop | 3-hop | 4-hop | Total |
|---|---|---|---|---|---|
| | 1 | 2,454 | 855 | 446 | 3,755 |
| | 2 | 2,425 | 853 | 467 | 3,745 |
| MQUAKE-CF | 3 | - | 827 | 455 | 1,282 |
| | 4 | - | - | 436 | 436 |
| | All | 4,879 | 2,535 | 1,804 | 9,218 |
| MQUAKE-T | 1 (All) | 1,421 | 445 | 2 | 1,868 |

Table 2: Data statistics of MQUAKE.

sure GPT-J can recall all unedited single-hop facts in the chains.

## 3.2 Data Construction of MQUAKE-T

Following a similar procedure to building MQUAKE-CF, we construct the other segment: MQUAKE-T, focusing on temporal-based, real-world fact updates. We take two dumps of Wikidata: `2021-04` and `2023-04`, and obtain the differences between the two versions. We find that most changes in Wikidata come from schema changes, i.e., (*Encyclopédie, instance of, encyclopedia → written work*) instead of actual fact updates. We then manually select 6 relations where the changes are most likely to align with real fact changes, e.g., (*United Kingdom, head of government, Boris Johnson → Rishi Sunak*). Similarly, we sample chains of facts and filter out unrecallable facts using GPT-J. When we generate edits given a fact chain, instead of sampling artificial counterfactual facts, we require that edits come from the diff set between the two versions of Wikidata. Note that different from

MQUAKE-CF, each instance in MQUAKE-T relates to only one edit, because all the edits are about position changes (e.g., *head of state*) and involving two in a question is not coherent. The goal of this dataset is to evaluate how successfully edited language models can answer questions which involve authentic updates to real-world knowledge.

## 3.3 Dataset Summary

**Dataset format** As shown in Table 1, each instance in the MQUAKE dataset is denoted by a tuple $d = \langle \mathcal{E}, \mathcal{Q}, a, a^*, \mathcal{C}, \mathcal{C}^* \rangle$, where $\mathcal{E}$ is a set of edits that we want to inject into the language model, $\mathcal{Q}$ represents multi-hop questions we use to evaluate editing methods (we provide three multi-hop questions), $a$ and $a^*$ denote the correct answer before and after edits, and $\mathcal{C}$ and $\mathcal{C}^*$ correspondingly represent the factual triples associated with this question before and after editing. A desirable knowledge editing method will inject all the edits in $\mathcal{E}$ into the model, and enable the model to internally use the edits and answer the questions.

**Data statistics** Table 2 summarizes the statistics of the MQUAKE-CF and MQUAKE-T datasets. The MQUAKE-CF dataset consists of more than 9K $N$-hop questions ($N \in \{2, 3, 4\}$), each of which associates with one or more edits.[3] We regard it as a diagnostic dataset to study the ability of edited models leveraging newly injected knowledge through editing methods. The MQUAKE-T dataset includes 1.8K instances, each of them associates with one real-world fact change.

**Number of edited facts** We consider two evaluation scenarios: a) First, we perform knowledge editing on only one instance $d$, which is associated with up to four edited facts. b) Then, we split the dataset into groups of $k$ instances ($k \in \{1, 100, 1000, 3000\}$ on MQUAKE-CF and $k \in \{1, 100, 500, 1868\}$ on MQUAKE-T), and consider all instances in a group at the same time and inject all the edited facts of these instances into the model at once. This harder setting is particularly interesting for editing methods such as MEMIT, which can handle large numbers of edits effectively (Meng et al., 2022b).

---

[3]Throughout the paper, our experiments on MQUAKE-CF are conducted on a randomly sampled subset of the full dataset which includes 3000 instances (1000 instances for each of {2,3,4}-hop questions) due to limited compute resources.

| Base Model | Method | Edit-wise | Instance-wise | Multi-hop | Multi-hop (CoT) |
|---|---|---|---|---|---|
| | Base | – | 100.0 | 43.4 | 42.1 |
| | FT | 44.1 | 24.1 | 1.6 ↓41.8 | 1.9 ↓40.2 |
| **GPT-J** | MEND | 72.8 | 59.6 | **9.2** ↓34.2 | 11.5 ↓30.6 |
| | ROME | 90.8 | 86.7 | 7.6 ↓35.8 | **18.1** ↓24.0 |
| | MEMIT | **97.4** | **94.0** | 8.1 ↓35.3 | 12.3 ↓29.8 |
| | Base | – | 61.0 | 30.0 | 36.6 |
| | FT | 20.2 | 7.8 | 0.7 ↓29.3 | 0.2 ↓36.4 |
| **Vicuna-7B** | MEND | 65.2 | 47.6 | 7.4 ↓22.6 | 8.4 ↓28.2 |
| | ROME | **99.8** | **89.6** | **8.4** ↓21.6 | **12.2** ↓24.4 |
| | MEMIT | 96.6 | 84.0 | 7.6 ↓22.4 | 9.0 ↓27.6 |

Table 3: Performance results on MQUAKE-CF (maximally 4 edits) for different knowledge editing methods using two base models, GPT-J and Vicuna-7B. We consider edits associated with each instance independently. Chain-of-thought (CoT) is included as a more advanced variant of prompt. *Base* denotes the model before editing. We include breakdown multi-hop (CoT) performance on MQUAKE-CF for {2,3,4}-hop questions and for questions with {1,2,3,4} edits in Appendix G.

## 3.4 Evaluation Metrics

We use the following metrics to measure whether the edits are made successfully. We include detailed formal definitions in Appendix B.

- **Edit-wise success rate** measures how many facts can be successfully recalled from the edited language model.

- **Instance-wise accuracy** measures in how many multi-hop instances, the model can recall all the individual single-hop facts. This is a reference metric for multi-hop performance, as the model must encode each individual fact in order to answer the multi-hop question. We measure instance-wise accuracy both before and after editing the model.

- **Multi-hop accuracy** measures the accuracy of the original and edited language models on multi-hop questions. In our datasets, there are three generated multi-hop questions for each instance. If any of the three questions is correctly answered by the model, we regard it as accurate. This is the main metric that we focus on to study models' ability to use edited knowledge consistently.

## 4 MQUAKE Challenges Model Editors

### 4.1 Experimental Setup

**Language models** We use GPT-J (6B) and Vicuna-7B (Chiang et al., 2023), which is a fine-tuned model based on LLaMA-7B (Touvron et al., 2023) as the baseline models to evaluate knowledge editing approaches. It is worth noting that existing parameter-update methods require access to a white-box language model and are very computationally expensive to apply to large models. In Section 5, we propose a lightweight approach, which can be applied to large black-box language models (Ouyang et al., 2022; Brown et al., 2020).

**Knowledge editing approaches** We evaluate the following state-of-the-art knowledge editing approaches on our datasets (more details can be found in Appendix C).

- **Fine-tuning (FT)** simply performs gradient descent on the edits to update model parameters. We follow Zhu et al. (2021) and fine-tune one layer in the model with a norm constraint on weight changes.

- **MEND** (Mitchell et al., 2022a) trains a hypernetwork to produce weight updates by transforming the raw fine-tuning gradients given an edited fact.

- **ROME** (Meng et al., 2022a) first localizes the factual knowledge at a certain layer in the Transformer architecture, and then updates the feedforward network in that layer to insert the new facts.

- **MEMIT** (Meng et al., 2022b) extends ROME to edit a large set of facts. It updates feedforward networks in a range of layers to encode all the facts.

Given an edited fact $(s, r, o \rightarrow o^*)$, we convert it to a cloze statement $t_r(s)$ and apply knowledge editing approaches with the objective of correctly predicting the edited object $o^*$ given $t_r(s)$. We

| Method | Edit-wise | Instance-wise | Multi-hop | Multi-hop (CoT) |
|--------|-----------|---------------|-----------|------------------|
| Base | – | 100.0 | 34.3 | 46.8 |
| FT | 19.5 | 19.0 | 0.0 ↓34.3 | 0.2 ↓46.6 |
| MEND | 99.0 | 98.5 | **16.0** ↓18.3 | **38.2** ↓8.6 |
| ROME | **100.0** | 97.7 | 0.3 ↓34.0 | 11.3 ↓35.5 |
| MEMIT | **100.0** | **98.9** | 0.3 ↓34.0 | 4.8 ↓42.0 |

Table 4: Performance results on MQUAKE-T for different knowledge editing methods using GPT-J as the base model. We consider edits associated with each instance independently. *Base* denotes the model before editing.

include the templates of cloze statement $t_r$ for each relation type $r$ in Appendix I.

**Evaluation metrics** As discussed in Section 3.4, we report **edit-wise** success rate, **instance-wise** accuracy, and **multi-hop** accuracy in our evaluation. We query the model with either manually-written prompt templates (for single-hop facts) or GPT-generated questions (for multi-hop fact chains). We adapt in-context learning and prompt the model with demonstrations when calculating instance-wise and multi-hop accuracy, in order to encourage the language model to recall and output knowledge in the desirable format (Brown et al., 2020).

We also consider chain-of-thought prompting (Wei et al., 2022) with in-context demonstrations to ensure the model's reasoning ability is fully utilized. See Appendix D for detailed prompts that we used to query language models. We denote the multi-hop accuracy with chain-of-thought prompting as **multi-hop (CoT)**.

## 4.2 Results on MQUAKE-CF

Table 3 shows the results on MQUAKE-CF when considering each instance individually across different methods with GPT-J and Vicuna-7B as the editing base models. As shown, all of the editing methods perform better than our fine-tuning baseline. In addition, they all gain traction on edit-wise evaluation, with MEMIT and ROME achieving higher than 90% accuracy with GPT-J and Vicuna-7B. In other words, when injecting a small number of edits, these techniques successfully inject the edits into language models and have the edited model recall them at inference time, which corroborates previous findings (Zhu et al., 2021; De Cao et al., 2021; Meng et al., 2022a; Mitchell et al., 2022a,b). Subsequently, a low edit-wise success rate entails a worse instance-wise accuracy (e.g., 59.6% for

MEND vs. 94.0% for MEND), as instance-wise correctness relies on recalling every fact from the model correctly for multi-hop questions.

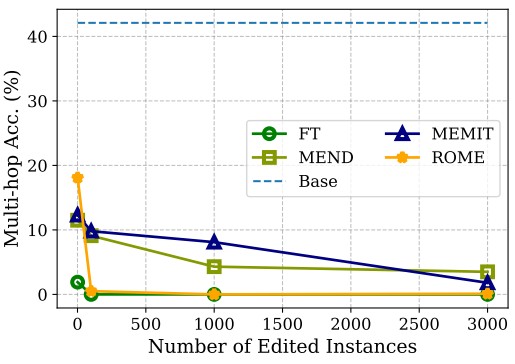

(a) MQUAKE-CF

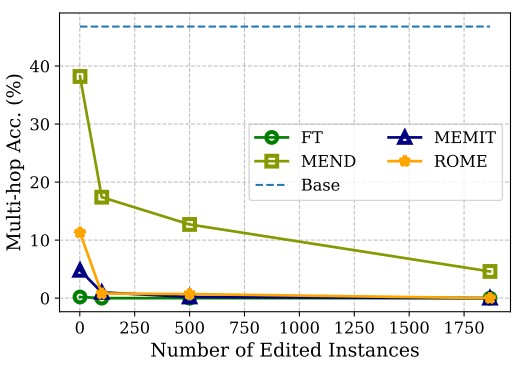

(b) MQUAKE-T

Figure 2: Multi-hop performance (CoT) of GPT-J before and after editing on (a) MQUAKE-CF and (b) MQUAKE-T across four different knowledge-editing methods with $k$ edited instances drawn for editing. $k \in \{1, 100, 1000, 3000\}$ on MQUAKE-CF. $k \in \{1, 100, 500, 1868\}$ on MQUAKE-T.

Surprisingly, the performance of edited models fails catastrophically at answering multi-hop questions. Even with the strongest baseline approach, MEMIT, multi-hop performance changes from $43.4\% \rightarrow 8.1\%$ with GPT-J and $30.0\% \rightarrow 7.6\%$ with Vicuna-7B. Our results lead to a surprising conclusion that, although these methods act faithfully when evaluating with single-hop questions, all of them fail catastrophically at answering multi-hop questions that rely on the edited facts. More importantly, compared to the ability to answer multi-hop questions prior to edits, model performance drops significantly as well. Our findings suggest that current knowledge-editing techniques, instead of integrating new facts into the model as new internal knowledge, are rather *hard coding* them into the model by updating weights locally. We hope these results can act as a call to the community to rethink the faithfulness of knowledge-editing methods and

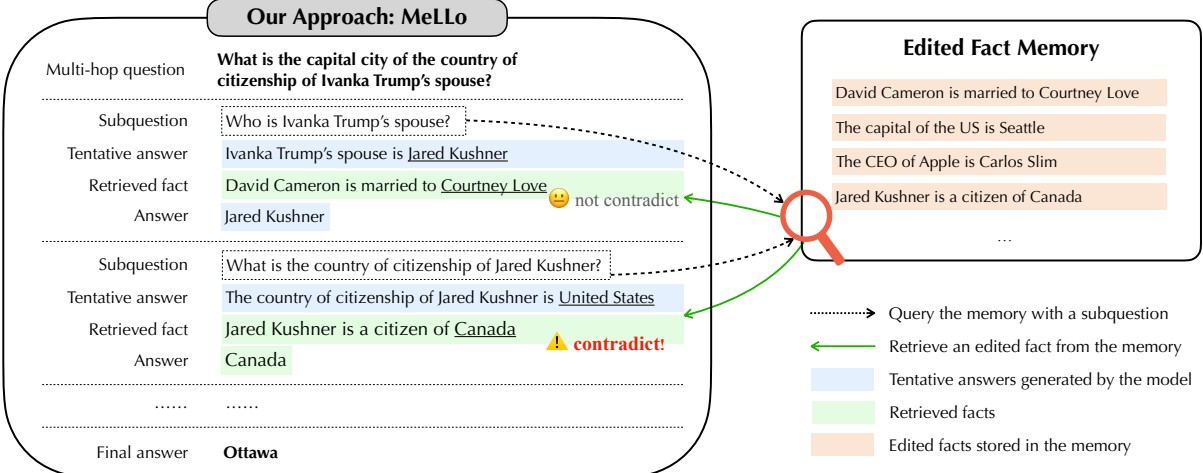

Figure 3: The illustration of our proposed method MeLLo. MeLLo decompose a multi-hop question into subquestions iteratively. When a subquestion is generated, the base model generates a tentative answer to the subquestion. Then, the subquestion is used to retrieve a most relevant fact from the edited fact memory. The model checks if the retrieved fact contradicts the generated answer and updates the prediction accordingly. The concrete prompts used in MeLLo are shown in Appendix F.

conduct deeper evaluations of edited models.

One hypothesis that these edited models cannot answer our multi-hop questions faithfully is that our prompt is not effective enough. Recent works suggest that providing explanations as Chain-of-thought (CoT) can greatly increase model performance even for models at the scale of 6B models (Wei et al., 2022). We further enhance our prompt with explanations and reevaluate all methods. Details about our CoT prompt template can be found in Appendix D. As shown in Table 3, CoT helps slightly across all settings yet still fails catastrophically at answering multi-hop questions. This further suggests that current knowledge-editing methods fail to update knowledge faithfully.

### 4.3 Results on MQUAKE-T

We evaluate all methods on GPT-J with real-world knowledge edit on MQUAKE.[4] The evaluation results are shown in Table 4. We find that in this setting, all methods except fine-tuning achieve near-perfect performance in terms of edit-wise and instance-wise accuracy. However, on multi-hop questions, the performance drops significantly compared to the base model before editing. We find that MEND works surprisingly well with CoT on MQUAKE-T. We hypothesize that this may be due to MEND being particularly effective in editing certain relations (e.g., *head of state*). On the

other hand, our results show that the edited model with CoT can substantially boost multi-hop performance. This suggests that explicit knowledge recall greatly helps the edited models answer multi-hop questions, while these models struggle to utilize the edited knowledge internally.

### 4.4 Evaluation with Edits at Scale

We extend our evaluation and consider all the edits from a randomly split group of $k$ instances at the same time ($k \in \{1, 100, 1000, 3000\}$ on MQUAKE-CF and $k \in \{1, 100, 500, 1868\}$ on MQUAKE-T). The results are shown in Figure 2. We find that, on both MQUAKE-CF and MQUAKE-T, the multi-hop performance of all methods further drops when injecting more edits into language models at the same time.

## 5 MeLLo: A Proposal for Editing Large Language Models

In Section 4, our evaluation results show that existing knowledge-editing methods fail catastrophically on multi-hop questions of MQUAKE. In this section, we present a simple but effective alternative, MeLLo (**Me**mory-based Editing for **L**arge **L**anguage Mo**dels).

### 5.1 Method

Figure 3 illustrates how MeLLo answers multi-hop questions. Inspired by memory-based knowledge-editing methods (Mitchell et al., 2022b), MeLLo

---

[4]We exclude Vicuna-7B on MQUAKE-T as it is trained more recently, and is likely to be contaminated with the new knowledge in our temporal questions.

keeps the base language model frozen and maintains all the edits in an explicit memory. During inference, MeLLo (1) decomposes a multi-hop questions into subquestions; (2) prompts the base language model to provide tentative answers to subquestions; and (3) self-checks whether the tentative answers contradict any edited facts in the memory. MeLLo can be applied easily to LLMs such as GPT-3 (Ouyang et al., 2022; Brown et al., 2020).

**Edited fact memory** MeLLo stores all the edited facts explicitly in memory. Specifically, all edited facts are first converted into sentence statements through manually-defined templates. Then, an off-the-shelf retrieval model (we use the pretrained `Contriever` model; Izacard et al. 2021) is used to embed all the edit statements and save them in a retrieval index. The index takes a query as input and returns an edited fact that is the most relevant (i.e., closest in the embedding space) to the query.

**Step-by-step generation and self-checking** To answer multi-hop questions with LLMs, we follow previous works and first prompt the model to decompose the multi-hop questions into multiple simple subquestions (Press et al., 2022; Zhou et al., 2023a). For example, in Figure 3, the first subquestion is *Who is Ivanka Trump's spouse?* Second, the model generates a tentative answer (e.g., *Jared Kushner*) to the subquestion based on the (unedited) knowledge stored in the model. Third, to assess whether the generated answer conflicts with any new knowledge edits, the subquestion is used as a query to retrieve a most relevant editing statement from the edited facts saved in memory. Fourth, the model is prompted to self-check if the retrieved fact contradicts the generated answer. If it does, the model adjusts the intermediate answer to this subquestion using the retrieved statement. Note that it is possible that a subquestion does not relate to any edited fact in memory as the corresponding fact is not edited; in this case, the model is prompted to keep the generated answer as the retrieved edit does not cause a contradiction. Finally, the model either generates the next subquestion of the multi-hop question or outputs the final answer.

## 5.2 Evaluation Results

We apply MeLLo on GPT-J (Wang and Komatsuzaki, 2021), Vicuna-7B (Chiang et al., 2023), and `text-davinci-003` (Ouyang et al., 2022; Brown et al., 2020). Table 5 shows performance of MeLLo on MQUAKE-CF and MQUAKE-T.

We find that with the same base model (i.e., GPT-J), MeLLo outperforms MEMIT and MEND significantly across all the settings while being more efficient and requiring no training. When incorporating MeLLo with a stronger LLM (`text-davinci-003`), MeLLo enlarges the performance gap substantially. This suggests that MeLLo works particularly well on strong base language models which can easily follow the instructions in our prompts. Along with its simplicity and efficacy, we think MeLLo can serve as a strong knowledge-editing baseline for future research. First, it does not require access to white-box model weights, so it is very extensible without any adaptation. Second, our base language model remains intact, avoiding the pitfall of overfitting to editing facts or destroying existing capacities due to weight updates. Third, we store edits in an explicit memory component instead of injecting facts into model parameters, which provides greater controllability in removing or adding knowledge on the fly.

We note that in order to answer multi-hop questions correctly after editing, the retriever we use in MeLLo needs to retrieve all the associated edited facts from the memory. In Appendix H, we investigate how retrieval accuracy affects the performance of MeLLo when using GPT-3 as the base model.

## 6 Related Work

**Knowledge-editing methods** Past work has investigated different approaches in editing LLMs at scale by injecting new knowledge into static model artifacts (Zhu et al., 2021; Sotoudeh and Thakur, 2019; Dai et al., 2022a; Hase et al., 2023; Zhou et al., 2023b; Dong et al., 2022; Huang et al., 2023). Some of these approaches include locating and modifying model weights that are responsible for specific concepts (Meng et al., 2022a,b; Dai et al., 2022b), and fast adaptation through a small auxiliary editing network (Mitchell et al., 2022a; De Cao et al., 2021). Recent work edits knowledge representations during decoding procedures of LLMs (Hernandez et al., 2023). Our proposed approach MeLLo share a similar spirit with SERAC (Mitchell et al., 2022b) where an explicit memory component is used to maintain all the edited facts. Different from SERAC, which trains additional models to incorporate the memory, MeLLo directly uses the base model to self-check whether the model generations need be adjusted. This allows MeLLo to be easily applied to black-

| | | MQUAKE-CF | | | | MQUAKE-T | | | |
|---|---|---|---|---|---|---|---|---|---|
| # Edited instances | | 1 | 100 | 1000 | 3000 | 1 | 100 | 500 | 1868 |
| Base Model | Method | | | | | | | | |
| GPT-J | MEMIT | 12.3 | 9.8 | 8.1 | 1.8 | 4.8 | 1.0 | 0.2 | 0.0 |
| GPT-J | MEND | 11.5 | 9.1 | 4.3 | 3.5 | 38.2 | 17.4 | 12.7 | 4.6 |
| GPT-J | MeLLo | 20.3 | 12.5 | 10.4 | 9.8 | 85.9 | 45.7 | 33.8 | 30.7 |
| Vicuna-7B | MeLLo | 20.3 | 11.9 | 11.0 | 10.2 | 84.4 | 56.3 | 52.6 | 51.3 |
| GPT-3 | MeLLo | **68.7** | **50.5** | **43.6** | **41.2** | **91.1** | **87.4** | **86.2** | **85.5** |

Table 5: Performance results of MeLLo (ours) on MQUAKE-CF and MQUAKE-T with GPT-J, Vicuna-7B, or GPT-3 (text-davinci-003) as the base language model. We consider a batch of $k$ instances as once ($k \in \{1, 100, 1000, 3000\}$ on MQUAKE-CF and $k \in \{1, 100, 500, 1868\}$ on MQUAKE-T). We include the best results with GPT-J from existing methods (MEMIT for MQUAKE-CF and MEND for MQUAKE-T) for comparison.

box LMs without any extra training.

**Knowledge-editing evaluation** The evaluation metrics for knowledge-editing techniques often involve verifying the updated answers by querying the edited facts or related facts (paraphrased or logically-entailed facts), as well as verifying that irrelevant facts are not corrupted (Meng et al., 2022a; Mitchell et al., 2022a; De Cao et al., 2021; Zhu et al., 2021; Hase et al., 2023). More recent work takes a step forward by evaluating LLMs' abilities to make inferences based on injected facts (Onoe et al., 2023) (e.g., after learning iPhone is a smartphone, the model should also know iPhone can browse the internet), or measuring the absence of unintended side effects of model edits (Hoelscher-Obermaier et al., 2023). Complementary with existing evaluation tools, MQUAKE particularly focuses on assessing whether edited models can answer multi-hop questions where the answer should change as an entailed consequence, showing that existing approaches fail on those questions.

**Prompting methods for multi-hop QA** Since the debut of effective base models such as GPT-3, prompt-based methods combined with an optional retrieval module have become a popular approach in handling multi-step QA tasks (Press et al., 2022; Yao et al., 2023; Khattab et al., 2022). Recent work also seeks to combine external NLI modules to justify whether answers to prompt-based queries are able to handle reasoning-based QA questions (Mitchell et al., 2022c). Our method is similar but more generic since we rely on the LLM itself to perform NLI step-by-step before reaching the final answer.

## 7 Conclusion

In this work, we present a benchmark MQUAKE that assesses knowledge editing methods for language models via multi-hop questions. We find that although edited language models can effectively recall edited facts, they fail on multi-hop questions that are entailed consequences of the edits. We propose a simple but effective alternative, MeLLo, which significantly outperforms existing knowledge editing methods. MeLLo does not require any additional training and can be applied to large LMs such as GPT-3 (Brown et al., 2020). We hope our work can facilitate future research on developing faithful knowledge editing methods.

## Limitations

The limitations of our work are as follows.

- We mainly evaluate existing knowledge editing methods on GPT-J (Wang and Komatsuzaki, 2021) and Vicuna (Chiang et al., 2023). The efficacy of these methods on other LLMs remains less explored. Note that existing editing methods are very computationally expensive. We leave the evaluation on other models as future work.

- We demonstrate that MeLLo outperforms existing knowledge editing methods on models with > 6B parameters. As MeLLo relies on language models for question decomposition and self-checking, future work may study how MeLLo works with smaller models such as GPT-2 (Radford et al., 2019).

- Our proposed memory-based approach, MeLLo, while being very effective on the MQUAKE benchmark, requires manually

defined prompts to drive language models on new tasks. Although we believe MeLLo is easy to instantiate on different tasks, we acknowledge this limitation and leave the evaluation on other tasks as future work.

- The multi-hop questions in MQUAKE are automatically generated by ChatGPT, rather than being crafted by humans. Although MQUAKE-T already involves real knowledge changes, we posit that the use of human-authored questions could further align MQUAKE with the realistic applications of knowledge editing methods.

## Acknowledgements

We thank Dan Friedman, Tianyu Gao, Eric Mitchell, Mengzhou Xia, Howard Yen, Jiayi Geng for providing valuable feedback. This research is partially supported by an NSF CAREER award (IIS-2239290), a Sloan Research Fellowship, and Microsoft Azure credits through the "Accelerate Foundation Models Academic Research" Initiative. ZZ is supported by a JP Morgan Ph.D. Fellowship. CM is a CIFAR Fellow.

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

## A    Details of Dataset Construction

### A.1    Sampling Fact Chains from Wikidata

We collect chains of facts that contain $N = \{2, 3, 4\}$ triples from Wikidata. We adopt heuristic rules to ensure that the sampled fact triples are coherent and lead to natural questions. Specifically, we apply the following constraints when sampling fact chains from Wikidata. (1) The sampled chain does not involve a circle; (2) The sampled chain does not contain two triples that share the same relation type; (3) The triples with the object being a country can only appear in the last two hops of the chain; (4) The sampled chain contains up to three object types; (5) All triples with a person or location object are consecutive in the chain; (6) The subject entity associated with the relation *headquarters location* (P159) must be a company or an organization; (7) In all triples with the relation *capital* (P36), the subject has to be a country. To use these heuristic rules, we manually label the object types for each relation we consider. For example, the relation *head of state* (P35) corresponds to a person as the object.

### A.2    Filtering Unrecallable Facts with GPT-J

We filter out any chain of facts which contain at least one fact that cannot be recalled by GPT-J. For each relation type, we manually define a question template as well as 8 demonstration examples for in-context-learning. We use in-context-learning to ensure the model can capture the answer format from the context. Table 6 shows an example of the prompt we use to recall facts of the relation *developer* (P178). We include the question templates of all relations on MQUAKE in Appendix I.

### A.3    Generating Questions using ChatGPT

Given a chain of facts, we prompt ChatGPT (gpt-3.5-turbo) to automatically generate multi-hop questions. The prompt we used is shown in Table 7.

In Table 8, we show some randomly selected examples of the questions generated by ChatGPT on MQUAKE-CF. We select 3 instances from $2, 3, 4$−hop questions, each of which contains three generated questions. As shown, ChatGPT successfully transforms the chain of triples into grammatically correct questions. Although these multi-hop questions are synthetic, they are logically consistent with the flow of the triple chains. We believe

these generated questions are of sufficient quality for assessing the efficacy of knowledge-editing methods.

## B    Evaluation Metrics

We evaluate the editing results based on three evaluation metrics: edit-wise success rate, instance-wise accuracy, and multi-hop accuracy. Suppose we have edited a language model and obtain the edited model $f^*(\cdot)$.

**Edit-wise** success rate measures how many edited facts can be recalled by the edited language model. Given an edit $e = (s, r, o \rightarrow o^*)$, the editing success is defined as $\mathbb{1}[f^*(t_r(s)) = o^*]$. We take the averaged value of the all edits in the dataset and refer it as the edit-wise success rate metric.

**Instance-wise** accuracy measures how many instances are there where all the associated facts can be recalled by the language model (either the original or edited one). Given an instance $d = \langle \mathcal{E}, \mathcal{Q}, a, a^*, \mathcal{C}, \mathcal{C}^* \rangle$, the instance-wise accuracy *before editing* is defined as

$$\mathbb{1}\left[\bigwedge_{(s,r,o)\in\mathcal{C}} [f(t_r(s)) = o]\right],$$

and the instance-wise accuracy *post editing* is defined as

$$\mathbb{1}\left[\bigwedge_{(s,r,o)\in\mathcal{C}^*} [f^*(t_r(s)) = o^*]\right].$$

We report the averaged instance-wise accuracy in our evaluation.

**Multi-hop** accuracy measures the accuracy on multi-hop questions. We regard an instance being predicted correctly if any of the multi-hop questions are answered correctly by the language model. Given an instance $d = \langle \mathcal{E}, \mathcal{Q}, a, a^*, \mathcal{C}, \mathcal{C}^* \rangle$, the multi-hop accuracy *before editing* is defined as

$$\mathbb{1}\left[\bigvee_{q\in\mathcal{Q}} [f(q) = a]\right],$$

and the multi-hop accuracy *post editing* is defined as

$$\mathbb{1}\left[\bigvee_{q\in\mathcal{Q}} [f^*(q) = a^*]\right].$$

We report the averaged multi-hop accuracy in our evaluation.

```
(In-context-learning examples)
Q: Who is the developer of Telegram? A: Telegram FZ-LLC
Q: Who is the developer of Microsoft Windows? A: Microsoft
Q: Who is the developer of PlayStation 2? A: Sony Interactive Entertainment
Q: Who is the developer of iTunes? A: Apple Inc.
Q: Who is the developer of SR-71 Blackbird? A: Kelly Johnson
Q: Who is the developer of Moblin? A: Linux Foundation
Q: Who is the developer of Xbox 360? A: Microsoft
Q: Who is the developer of Kinsey scale? A: Alfred Kinsey
(Query during inference)
Q: Who is the developer of SteamOS? A: Valve Corporation
```

Table 6: An example of the prompt we used to recall single-hop factf with relation *developer* (P178) from language models. We use in-context-learning with 8 demonstration examples to ensure the model can output the answer in a desirable format.

```
System:
You are a powerful multi-hop question generator. Users will provide a chain of Wikidata triples,
and you will help write questions to ask the tail entity from the head entity.  You shouldn't
include bridge entities in generated questions. The questions should only include the head entity.

[12 in-context demonstrations abbreviated]

User:
Given Wikidata triples (Daredevil/Bullseye: The Target, author, x1), (x1, country of citizenship,
x2), (x2, continent, x3), write a question to ask x3. Don't mention x1, x2, ...  Write three
possible questions in natural English.

System:
1. What continent is the country of citizenship of the author of Daredevil/Bullseye: The Target
located in?
2. From which continent does the author of Daredevil/Bullseye: The Target's country of citizenship
belong?
3. What continent is the author's country of citizenship, who wrote Daredevil/Bullseye: The Target,
situated in?
```

Table 7: An example of using ChatGPT (gpt-3.5-turbo) to generate questions from Wikidata triples. We manually write 12 demonstrations as the prompt when querying ChatGPT.

## C   Implementation Details for Knowledge Editing Methods

### C.1   Fine-tuning

Our fine-tuning baseline (FT) performs gradient descent on the edits to update model parameters. We fine-tune layer 21 of GPT-J and layer 31 of Vicuna-7B. We follow Zhu et al. (2021) and use a norm constraint on weight changes with a coefficient $5 \times 10^{-5}$ in our implementation.

### C.2   MEND

We use the GPT-J MEND editor trained by Meng et al. (2022a). For Vicuna-7B, we train our own MEND editor model on the Wikitext generation editing dataset (Mitchell et al., 2022a) with the different hyperparameters. During inference, we set the learning rate scale to be 1.0.

### C.3   ROME

For GPT-J, we use the default hyperparameters of ROME and the pre-computed covariance statistics released by Meng et al. (2022a). For Vicuna-7B, we run ROME to update model weights at layer 9 with the default hyperparameters. We compute the covariance statistics for Vicuna-7B on Wikitext using a sample size of 100,000.

### C.4   MEMIT

For GPT-J, we use the default hyperparameters of MEMIT and the pre-computed covariance statistics released by Meng et al. (2022b). For Vicuana-7B, we update model weights at layers $\{5, 6, 7, 8, 9\}$ with the default hyperparameters. Similarly, we compute the covariance statistics for Vicuna-7B on Wikitext using a sample size of 100,000.

| | **Examples of 2-hop questions** |
|---|---|
| $\mathcal{C}$ | (Jacques Necker, employer, University of Geneva) (University of Geneva, headquarters location, Geneva) |
| $\mathcal{Q}$ | What is the location of the headquarters of the employer of Jacques Necker? |
| | Where is the employer of Jacques Necker headquartered? |
| | In which city is the head office located for the company that employed Jacques Necker? |
| $\mathcal{C}$ | (Percival Lowell, educated at, Harvard University) (Harvard University, headquarters location, Cambridge) |
| $\mathcal{Q}$ | What is the location of the headquarters of the institution where Percival Lowell was educated? |
| | In which city is the institution located where Percival Lowell received his education? |
| | Where is the headquarters of the educational institution attended by Percival Lowell located? |
| $\mathcal{C}$ | (Gordon Moore, country of citizenship, United States of America) (United States of America, capital, Washington, D.C.) |
| $\mathcal{Q}$ | What is the capital of the country where Gordon Moore holds citizenship? |
| | Which is the capital city of the country to which Gordon Moore belongs? |
| | In which city is the seat of the government of the country where Gordon Moore is a citizen? |
| | **Examples of 3-hop questions** |
| $\mathcal{C}$ | (Kim Kardashian, spouse, Kanye West) (Kanye West, genre, hip hop music) |
| | (hip hop music, country of origin, United States of America) |
| $\mathcal{Q}$ | What is the country of origin of the genre associated with the spouse of Kim Kardashian? |
| | From which country does the genre of the partner of Kim Kardashian hail? |
| | Which country is the genre of the partner of Kim Kardashian associated with originally from? |
| $\mathcal{C}$ | (Nicholas of Tolentino, religion or worldview, Catholic Church) (Catholic Church, founded by, Jesus Christ) |
| | (Jesus Christ, place of birth, Bethlehem) |
| $\mathcal{Q}$ | Where was the founder of Nicholas of Tolentino's religion born? |
| | In which city was the founder of the religion that Nicholas of Tolentino adhered to born? |
| | What is the birthplace of the founder of the religion that Nicholas of Tolentino followed? |
| $\mathcal{C}$ | (Boston, head of government, Marty Walsh) (Marty Walsh, educated at, Boston College) |
| | (Boston College, headquarters location, Chestnut Hill) |
| $\mathcal{Q}$ | In what city is the headquarters of the institution where the head of government of Boston was educated located? |
| | Where is the location of the headquarters of the educational institution where the head of government of Boston received their education? |
| | What is the city where the headquarters of the institution where the head of government of Boston was educated at located? |
| | **Examples of 4-hop questions** |
| $\mathcal{C}$ | (Xbox Live, developer, Microsoft) (Microsoft, chief executive officer, Satya Nadella) |
| | (Satya Nadella, place of birth, Hyderabad) (Hyderabad, continent, Asia) |
| $\mathcal{Q}$ | Which continent is home to the birthplace of the CEO of Xbox Live developer? |
| | Where was the CEO of the developer of Xbox Live born in which continent? |
| | In what continent was the CEO of Xbox Live's developer born? |
| $\mathcal{C}$ | (Winnie the Pooh, creator, A. A. Milne) (A. A. Milne, child, Christopher Robin Milne) |
| | (Christopher Robin Milne, country of citizenship, United Kingdom) (United Kingdom, official language, English) |
| $\mathcal{Q}$ | What is the official language of the country where the child of Winnie the Pooh's creator holds citizenship? |
| | Which language is officially spoken in the country where the child of the creator of Winnie the Pooh is a citizen? |
| | What is the officiated language of the country where the child of Winnie the Pooh's creator is a citizen of? |
| $\mathcal{C}$ | (watchOS, developer, Apple Inc.) (Apple Inc., chief executive officer, Tim Cook) |
| | (Tim Cook, country of citizenship, United States of America) (United States of America, capital, Washington, D.C.) |
| $\mathcal{Q}$ | What is the capital of the country where the CEO of the developer of watchOS holds citizenship? |
| | In which city does the CEO of the company that developed watchOS hold citizenship? |
| | Which city is the capital of the home country of the CEO of the developer of watchOS? |

Table 8: Qualitative examples of the generated multi-hop questions on MQUAKE-CF. Given a chain of facts, we query ChatGPT (gpt-3.5-turbo) to generate multi-hop questions with the prompt shown in Table 7.

## D Chain-of-thought Prompting for Multi-hop Questions

We use chain-of-thought (CoT) prompting (Wei et al., 2022) to maximize model performance. Table 9 shows one simplified example of our prompt with CoT.

## E Extended Golden Labels for MQUAKE-T

Our MQUAKE-T contains limited test cases. To better assess the model's original performance on multi-hop questions, we extend the possible golden labels for each multi-hop question. Specifically, we allow outdated answers given smaller language

```
Question: What is the capital of the country where Plainfield Town Hall is located?
Thoughts: Plainfield Town Hall is located in the country of the United States of America. The
capital of United States is Washington, D.C.
Answer: Washington, D.C.

Question: In which country is the company that created Nissan 200SX located?
Thoughts: Nissan 200SX was created by Nissan. Nissan is located in the country of Japan.
Answer: Japan

[3 in-context demonstrations abbreviated]

Question: Who has ownership of the developer of the Chevrolet Corvette (C4)?
Thoughts: The developer of Chevrolet Corvette (C4) is Chevrolet. Chevrolet is owned by General
Motors.
Answer: Model Generated Answer Goes Here
```

Table 9: The template of the prompt we used for asking multi-hop questions using chain-of-thoughts.

```
Please answer the following question faithfully using the knowledge you have from Wikipedia.
Provide 10 possible answers to the question, using all the Wikipedia data you know. Rank them from
the most current to the most outdated.

Input: What is the name of the current head of the United States of America government?
Output:
Joe Biden
Donald Trump
Barack Obama
George W. Bush
Bill Clinton
George H. W. Bush
Ronald Reagan
Jimmy Carter
Gerald Ford
Richard Nixon

Input: What is the name of the current head of the New York City government?
Output: Model Generated Answer Goes Here
```

Table 10: The template of the prompt we used for extending golden labels for MQUAKE-T. The prompt contains one demonstration for better aligning model behaviors.

models tend to be less calibrated. To extend the golden labels, we use GPT-3 (text-davinci-003) to query outdated answers. See the prompt we used in Table 10.

## F  Prompts used in MeLLo

The prompt we used in MeLLo is shown in Table 11. We first prompt the language model to decompose subquestions. Then the language model generates a tentative question to the subquestion (marked in green text); then we use the generated subquestion to retrieve the most relevant edited fact (marked in light blue text) and append it to the prompt. The model self-checks if the retrieved fact contradicts the generated answer. The prompting procedure goes iteratively until the model generates the final answer.

## G  Breakdown Results on MQUAKE-CF

Table 12 and Table 13 present the breakdown results on MQUAKE-CF when using GPT-J as the base model. We find that, in all editing methods (1) the performance on 2-hop questions is much higher than 3-hop and 4-hop questions; (2) the performance is worse when there are more edits asssociated with the edited instances.

## H  Impact of Retrieval Performance

In MeLLo, in order to answer multi-hop questions correctly after editing, the retrieval model needs to retrieve all the associated edited facts (each question is associated with 1-4 edited facts) from the memory. Here we investigate how retrieval accuracy affects the performance of MeLLo when using GPT-3 as the base model. We compute the retrieval accuracy (i.e., how many instances where all the as-

```
[4 in-context demonstrations abbreviated]

Question: What is the capital city of the country of citizenship of Ivanka Trump's spouse?
Subquestion: Who is Ivanka Trump's spouse?
Generated answer: Ivanka Trump's spouse is Jared Kushner.
Retrieved fact: David Cameron is married to Samantha Cameron.
Retrieved fact does not contradict to generated answer, so the intermediate answer is: Jared
Kushner
Subquestion: What is the country of citizenship of Jared Kushner?
Generated answer: The country of citizenship of Jared Kushner is United States.
Retrieved fact: Jared Kushner is a citizen of Canada.
Retrieved fact contradicts to generated answer, so the intermediate answer is: Canada
Subquestion: What is the capital city of Canada?
Generated answer: The capital city of Canada is Ottawa.
Retrieved fact: The capital city of United States is Seattle.
Retrieved fact does not contradict to generated answer, so the intermediate answer is: Ottawa
Final answer: Ottawa
```

Table 11: A step-by-step illustration of MeLLo solving one simplified example. Green parts are generated by the language model, and blue parts are facts retrieved by the retriever.

|  | 2-hop | 3-hop | 4-hop | All |
|---|---|---|---|---|
| **Base** | 47.5 | 27.1 | 45.3 | 42.1 |
| **FT** | 3.7 | 1.4 | 0.5 | 1.9 |
| **MEND** | 13.9 | 11.3 | 9.5 | 11.5 |
| **ROME** | 33.8 | 9.1 | 11.4 | 18.1 |
| **MEMIT** | 22.5 | 6.0 | 8.4 | 12.3 |

Table 12: Breakdown multi-hop performance (CoT) on MQUAKE-CF for {2,3,4}-hop questions. We use GPT-J as the base model in this experiment

| # Edits = | 1 | 2 | 3 | 4 | All |
|---|---|---|---|---|---|
| **Base** | 34.0 | 43.0 | 40.4 | 51.7 | 42.1 |
| **FT** | 4.2 | 0.7 | 0.3 | 0.0 | 1.9 |
| **MEND** | 16.0 | 11.0 | 7.3 | 4.4 | 11.5 |
| **ROME** | 23.8 | 20.9 | 9.0 | 2.6 | 18.1 |
| **MEMIT** | 20.5 | 9.8 | 5.5 | 2.6 | 12.3 |

Table 13: Breakdown multi-hop performance (CoT) on MQUAKE-CF for questions with {1,2,3,4} edits. We use GPT-J as the base model in this experiment.

| # Instances ($k =$) | 1 | 100 | 1000 | 3000 |
|---|---|---|---|---|
| **Retrieval acc.** | 93.6 | 67.7 | 59.4 | 58.7 |
| **MeLLo (on GPT-3)** | 68.7 | 50.5 | 43.6 | 41.2 |

Table 14: How retrieval accuracy affects the multi-hop performance of MeLLo on MQUAKE-CF. We consider a group of $k$ instances at the same time. *Retrieval acc.*: how many instances where all the associated edited facts are correctly retrieved from the memory.

sociated edited facts are correctly retrieved from the memory) when applying MeLLo on MQUAKE-

CF with GPT-3 by considering different numbers of edited instances at the same time. As Table 14 shows, the performance of MeLLo decreases if the retrieval accuracy is lower (as a result of considering more instances at the same time). Among those questions where all associated facts are successfully retrieved from memory, MeLLo can answer 73.1% of them correctly. This indicates that retrieval performance can significantly impact the model performance. When we consider more irrelevant knowledge edits in the memory, retrieval can be more challenging, and we expect that using more advanced retrieval techniques can improve the performance.

# I  Question/Cloze Statement Templates used in MQUAKE

Table 15 shows the question templates and the cloze-style statement templates we use in MQUAKE. We use the question templates to query single-hop facts when we use GPT-J for filtering and use the cloze-style statement templates to convert an edited fact to a natural language statement.

| Relation | Question template | Cloze-style statement template |
|---|---|---|
| P30 | Which continent is [S] located in? | [S] is located in the continent of |
| P36 | What is the capital of [S]? | The capital of [S] is |
| P35 | What is the name of the current head of state in [S]? | The name of the current head of state in [S] is |
| P6 | What is the name of the current head of the [S] government? | The name of the current head of the [S] government is |
| P20 | Which city did [S] die in? | [S] died in the city of |
| P26 | Who is [S] married to? | [S] is married to |
| P140 | Which religion is [S] affiliated with? | [S] is affiliated with the religion of |
| P1412 | What language does [S] speak? | [S] speaks the language of |
| P19 | Which city was [S] born in? | [S] was born in the city of |
| P69 | Which university was [S] educated at? | The univeristy where [S] was educated is |
| P40 | Who is [S]'s child? | [S]'s child is |
| P27 | What is the country of citizenship of [S]? | [S] is a citizen of |
| P175 | Who performed [S]? | [S] was performed by |
| P108 | Who is the employer of [S]? | [S] is employed by |
| P112 | Who founded [S]? | [S] was founded by |
| P50 | Who is the author of [S]? | The author of [S] is |
| P170 | Who was [S] created by? | [S] was created by |
| P407 | Which language was [S] written in? | [S] was written in the language of |
| P37 | What is the official language of [S]? | The official language of [S] is |
| P740 | Where was [S] founded? | [S] was founded in the city of |
| P495 | Which country was [S] created in? | [S] was created in the country of |
| P106 | What kind of work does [S] do? | [S] works in the field of |
| P136 | What type of music does [S] play? | The type of music that [S] plays is |
| P364 | What is the original language of [S]? | The original language of [S] is |
| P937 | Which city did [S] work in? | [S] worked in the city of |
| P800 | What is [S] famous for? | [S] is famous for |
| P641 | Which sport is [S] associated with? | [S] is associated with the sport of |
| P413 | What position does [S] play? | [S] plays the position of |
| P286 | Who is the head coach of [S]? | The head coach of [S] is |
| P159 | Which city is the headquarter of [S] located in? | The headquarters of [S] is located in the city of |
| P178 | Who is the developer of [S]? | [S] was developed by |
| P488 | Who is the chairperson of [S]? | The chairperson of [S] is |
| P169 | Who is the chief executive officer of [S]? | The chief executive officer of [S] is |
| P449 | Who is the original broadcaster of [S]? | The origianl broadcaster of [S] is |
| P176 | Which company is [S] produced by? | The company that produced [S] is |
| P1037 | Who is the director of [S]? | The director of [S] is |
| P1308 | Who is the [S]? | The [S] is |

Table 15: Question templates and cloze-style statement templates that are used in MQUAKE. "[S]" represents a placeholder for the subject entity of the fact. We use the question templates to query single-hop facts when we use GPT-J for filtering and use the cloze-style statement templates to convert an edited fact to a statement.