# OpenReview forum: "MQuAKE: Assessing Knowledge Editing in Language Models via Multi-Hop Questions"
_EMNLP/2023/Conference — EMNLP 2023 Main_

### Official Review · Reviewer_ipS4 · 2023-08-02

**Soundness:** 4

**Excitement:**

4: Strong: This paper deepens the understanding of some phenomenon or lowers the barriers to an existing research direction.

**Paper Topic And Main Contributions:**

The paper addresses the problem of knowledge editing in large language models without changing the LLM weights. Authors present a benchmark MQuAKE to assess the ability of the edited model to answer multi-hop questions with the correct answer depending on the edited information. Also, the authors propose an external memory-based method MeLLo for knowledge-editing without changing the target model weights.

**Reasons To Accept:**

The paper has a clear message and proposes a new benchmark for the task that is of interest to the community and also proposes a new lightweight baseline approach for knowledge editing of LLMs. The dataset construction and experiments are well-described. Limitations show a good cover of possible future work directions.

**Reasons To Reject:**

Prompt templates for the main experiments mentioned in lines 387-389 were not provided in the paper.

**Reproducibility:**

5: Could easily reproduce the results.

**Reviewer Confidence:**

3: Pretty sure, but there's a chance I missed something. Although I have a good feel for this area in general, I did not carefully check the paper's details, e.g., the math, experimental design, or novelty.

---

> ### Author Rebuttal · Authors · 2023-08-29
>
> Thank you for your valuable feedback! We are encouraged that you find our paper has a clear message, proposes a new benchmark, and proposes a new approach MeLLo without changing the target model weights!
>
> **“Prompt templates for the main experiments mentioned in lines 387-389 were not provided in the paper”**
>
> Thanks for pointing this out! We will include templates we defined for each relation as well as the in-context demonstrations we used to query multi-hop questions in the appendix in the revision. Here we provide a few prompt templates as examples.
>
> | Relation                     | Prompt template                            |
> |------------------------------|--------------------------------------------|
> | developer (P178)             | Who is the developer of [X]?               |
> | country of citizenship (P27) | What is the country of citizenship of [X]? |
> | educated at (P69)            | Which university was [X] educated at?      |
> | owner (P127)                 | Who owns [X]?                              |
> | … (33 more relations)        | …                                          |

---

### Official Review · Reviewer_i7oa · 2023-08-05

**Soundness:** 4

**Excitement:**

4: Strong: This paper deepens the understanding of some phenomenon or lowers the barriers to an existing research direction.

**Paper Topic And Main Contributions:**

This paper explores consistency in knowledge editing. It introduces the concept of "chains-of-facts" to investigate the impact of a particular piece of knowledge on the entire chain of edits. Specifically, two datasets, mquake-t and mquake-cf, are constructed for GPT-J. By formulating multi-hop questions related to chains-of-facts, the paper aims to determine whether the model has truly learned new knowledge or simply hard-coded it. The article presents a benchmark that challenges traditional knowledge editing methods and also proposes a memory-based approach called mello that effectively addresses the editing problem with chains-of-facts.

**Reasons To Accept:**

1. The writing in is very clear and the motivation is strong.
2. Previous knowledge editing primarily focused on whether the model can recall the injected new knowledge and its paraphrases. However, this paper proposes chains-of-facts, which provides a new perspective for exploring whether the model truly learns the injected knowledge.
3. This paper designs various evaluation metrics for chains-of-thoughts and also considers different prompting methods such as in-context learning and chain-of-thoughts.
4. To address the identified issues, this paper proposes a memory-based approach that effectively applies to chains-of-facts and can be applicable to models as large as 175B or even larger.

**Reasons To Reject:**

Overall, this paper does not have any glaring flaws that would lead to rejection. However, there are areas where the design of the dataset and experiments could be improved:

1. The world knowledge of different models varies, and although the authors consider GPT-J, Vicuna, and GPT-3, the dataset proposed in this paper is still limited to GPT-J. On one hand, the authors filter out facts that cannot be recalled by GPT-J. On the other hand, the construction of Mquake-t's temporal segments is also based on GPT-J's design. This limits the applicability of the dataset to a very narrow range of language models.

2. The paper primarily uses accuracy as an evaluation metric, but because objects can have aliases, it would be better to use a probability-based approach like CounterFact (ROME) to assess it.

3. The proposed method, Mello, employs a memory-based approach, and the accuracy and recall rate of the retriever will have a significant impact on the language model's output. The current dataset setting may be too simplistic, allowing the retriever to effectively retrieve relevant factual edits. It is important to investigate the potential impact on the final output if the retriever retrieves irrelevant knowledge. I am curious about the robustness of Mello in handling retriever errors in recalling facts.

**Reproducibility:**

5: Could easily reproduce the results.

**Reviewer Confidence:**

5: Positive that my evaluation is correct. I read the paper very carefully and I am very familiar with related work.

---

> ### Author Rebuttal · Authors · 2023-08-29
>
> Thank you for the valuable comments! We are glad that you find that our paper writing is very clear and the motivation is strong, our proposed datasets MQuAKE and evaluation metrics provide a new perspective for understanding the injected knowledge, and our proposed method MeLLo is effective and is applicable to models as large as 175B. We have addressed your detailed comments below, and we are more than willing to engage in further discussion should any follow-up questions arise!
>
> **“The dataset proposed in this paper is limited to GPT-J”**
>
> We acknowledge that this method of data construction does give a bias towards GPT-J. We argue that (1) the filtering process is very important, as it increases the probability of the multi-hop questions being meaningful and answerable by language models (otherwise, the accuracy will be too low, which is hard to draw conclusions); (2) we chose GPT-J as the model for filtering because it is fairly small and uses similar data to most other general-purpose model (so it’s a good approximate filter for other models), and GPT-J is a common base model that is studied in prior knowledge editing works.
>
> We note that MQuAKE *can* also be applied to other models -- one can compare before-editing performance and post-editing performance. In addition, the proposed data construction pipeline can be easily applied with other models as the filter.
>
>
> **“It would be better to use a probability-based approach like CounterFact (ROME) to assess it”**
>
> We chose to use accuracy instead of probability-based metric because accuracy aligns better with the end use of the language models (e.g., when we ask a question to LMs, we expect them to return the answers instead of only assigning higher probability to the answers). We also acknowledge that there may be multiple forms of answers to some questions -- we leverage aliases from Wikidata to mitigate this issue.
>
> **“The accuracy and recall rate of the retriever will have a significant impact on the language model's output”**
>
> Thanks for the great question, we think this is an excellent point to highlight! We have conducted new analyses to answer the reviewer's question.
>
> Yes, the performance of MeLLo is significantly impacted by the retrieval performance. In order to answer the multi-hop questions after editing, the retriever needs to retrieve **all** the associated edited facts (each question is associated with 1-4 edited facts) from memory.
>
> To investigate the effect of retrieval performance, we compute the retrieval accuracy (i.e., how many edited facts are correctly retrieved from the memory) when applying MeLLo on MQuAKE-CF with GPT-3 by considering different numbers of edited instances at the same time. As the table below shows, the performance of MeLLo decreases if the retrieval accuracy is lower (as a result of considering more instances at the same time). Among those questions where all associated facts are successfully retrieved from memory, MeLLo can answer 73.1% of them correctly. These empirical results verify that retrieval performance can significantly impact the model performance. When we consider even more irrelevant knowledge in the memory, retrieval might be more challenging, and we expect using more advanced retrieval techniques can improve the performance. We will include this analysis in the revised paper.
>
> | # Edited instances | 1    | 100  | 1000 | 3000 |
> |--------------------|------|------|------|------|
> | Retrieval accuracy | 93.6 | 67.7 | 59.4 | 58.7 |
> | MeLLo (on GPT-3)   | 68.7 | 50.5 | 43.6 | 41.2 |

---

### Official Review · Reviewer_ojrC · 2023-08-10

**Soundness:** 4

**Excitement:**

3: Ambivalent: It has merits (e.g., it reports state-of-the-art results, the idea is nice), but there are key weaknesses (e.g., it describes incremental work), and it can significantly benefit from another round of revision. However, I won't object to accepting it if my co-reviewers champion it.

**Paper Topic And Main Contributions:**

This paper presents a dataset for evaluating knowledge editing in LMs, focusing on evaluating multi-hop questions whose answers should change as the result of and edit to the answer to one of its sub questions. The authors construct two versions of this dataset from WikiData, where multi-hop questions are constructed from knowledge-base tuples using ChatGPT. Their first dataset is larger, consisting of chains of 37 different relation types up to a length of 4 connecting the 20% of most popular entities in WikiData. Relations are further filtered based on whether GPT-J is able to recall each individual fact. To construct edits, the authors replace the object of a subject-relation-object tuple with another object sharing the same relation. The authors also construct a smaller dataset containing real factual edits that have occurred over time (2021 to 2023), focusing on changes in 6 different relation types.

The authors experiment with editing in a single or multiple (up to 3K) facts into a single LM and experiment with several baselines. They find that existing methods, while they're able to recall individual edited facts, they are unable to successfully use these updated facts to answer multi-hop questions using them, failing catastrophically. When allowed to perform chain-of-thought reasoning for multi-hop questions, however, all methods are able improve and recover much of their base performance. This observation holds true for both versions of their dataset.

The authors finally propose their own method for knowledge editing, that is based around (1) decomposing multi-hop questions into their subquestions and (2) performing retrieval for each subquestion from a non-parametric corpus of edited facts. The LM then makes a decision whether the answer should be affected by this retrieved fact, then updates its answer based on that decision. The authors demonstrate that this pipeline outperforms

**Reasons To Accept:**

The paper is well-motivated, highlighting a overlooked issue knowledge editing work.

The dataset and its construction process may support future work in this area.

The paper is well-written and includes a variety of knowledge-editing baselines and base LMs in experiments.

**Reasons To Reject:**

While this dataset provides a method for evaluating whether facts are successfully edited in multi-hop questions, it fails to also evaluate the specificity/locality of edited facts and whether edited facts are having unintended effects (e.g., catastrophic forgetting) elsewhere. Evaluating the specificity of edits is a consistent concern in other works proposing datasets for knowledge editing. For instance, this dataset does not consider questions such as whether editing (WALL-E, creator, Andrew Stanton → James Watt) affects other facts, for instance (Finding Nemo, creator, Andrew Stanton → James Watt).

The baselines and comparisons for the proposed method in Section 5 make it unclear where improvements in the proposed method are coming from. In particular, some of the cited work has demonstrated that, for multi-hop QA, CoT prompting  by decomposing inputs into QA pairs outperforms standard paragraph-style CoT prompting. Baselines where edited models receive the same prompt, with the retrieval and counterfactual reasoning steps removed, would've been well-placed here.

Adding onto the point above, some of the cited work (SERAC) has proposed a method that similarly (1) performs retrieval over edited facts, (2) determines whether the edited facts should affect the answer to the given question and (3) use the edited fact to produce the correct output. The primary difference here is the usage of a single LM to do all these inferences jointly rather than training separate components. Comparisons between these two methods of performing steps 1-3 would also add context for the proposed method.



**Reproducibility:**

4: Could mostly reproduce the results, but there may be some variation because of sample variance or minor variations in their interpretation of the protocol or method.

**Reviewer Confidence:**

4: Quite sure. I tried to check the important points carefully. It's unlikely, though conceivable, that I missed something that should affect my ratings.

---

> ### Author Rebuttal · Authors · 2023-08-29
>
> We appreciate your valuable feedback! We are encouraged that you find that our paper is well-motivated, our datasets MQuAKE and the construction process may support future research, and our experiments include a variety of baselines and base LMs. We have addressed your detailed comments below. Additionally, we will incorporate all the new results in the response into the revised version.
>
> **“The dataset fails to also evaluate the specificity/locality of edited facts and whether edited facts are having unintended effects”**
>
> Excellent point. MQuAKE (like all benchmarks) should be used in conjunction with other evaluation tools and resources. Our paper particularly focuses on assessing whether edited models can answer questions where the answer should change as an entailed consequence, showing that existing approaches fail on those questions. We agree that evaluating specificity is essential for knowledge editing methods and we plan to add it in our revision. We’d also like to point out that there are already results for the specificity of all the baseline methods (but not MeLLo) in previous studies (e.g., Meng et al., 2022).
>
> **“It's unclear where improvements in the proposed method are coming from”**
>
> Good point! To address this, we ran new experiments testing the baselines by using the same QA-style prompt without retrieval and self-checks. This experiment is conducted on GPT-J by considering 3000 instances from MQuAKE-CF or 1868 instances from MQuAKE-T at the same time. As the table below shows, we do not observe substantial improvements on baselines by simply changing the prompts. It is likely because of the limited capacity of performing QA-style CoT for those small-sized models (i.e., GPT-J). This suggests that the improvements mainly come from the editing methods instead of using QA-style prompts.
>
> |            | MQuAKE-CF (3000 instances) |                | MQuAKE-T (1868 instances) |                |
> |------------|----------------------------|----------------|---------------------------|----------------|
> | **Method** | CoT (standard)             | CoT (QA-style) | CoT (standard)            | CoT (QA-style) |
> | MEMIT      | 1.8                        | 2.1            | 0.0                       | 0.0            |
> | MEND       | 3.5                        | 3.5            | 4.6                       | 5.7            |
> | MeLLo      | -                          | 9.8            | -                         | 30.7           |
>
> We note that it is possible that performance gains of using QA-style are larger when using a stronger base model (e.g., GPT-3). However, existing methods are not applicable to large-scale or black-box models. We leave further investigation on large models as future work.
>
> **“Differences between MeLLo and SERAC?”**
>
> Instead of training a separate classifier model and a counterfactual model, MeLLo directly uses the LLM to self-check whether the retrieved fact contradicts the model knowledge or not. MeLLo can be easily applied to black-box LMs without any extra training (as shown in the experiments on GPT-3), which is an advantage over existing editing methods including SERAC. With the same base model (e,g, GPT-3), the point-to-point performance comparison between MeLLo and SERAC will highly depend on the quality of the classifier model and the counterfactual model that SERAC trained, which we leave as future work.

---

### Official Review · Reviewer_uapW · 2023-08-11

**Soundness:** 4

**Excitement:**

4: Strong: This paper deepens the understanding of some phenomenon or lowers the barriers to an existing research direction.

**Missing References:**

Given that the majority of my experience with multi-hop QA comes from more of an IR perspective, a lot of the references I have in mind don't seem to be applicable here.

**Paper Topic And Main Contributions:**

This work aims to find additional ways to evaluate knowledge-edited LLMs, in particular by evaluating how the edited facts affect multi-hop questions due to a ripple-like effect this would have on the paths the language model would have to take in order to retrieve/generate the correct response after the edit. The main contribution is the benchmark they provide, MQuAKE, as well as a novel method for knowledge editing, MeLLo. The authors demonstrate the difficulty models have at solving MQuAKE, suggesting that prior methods for knowledge editing are limited in how they can enable language models to recall facts in the newly edited world.

**Questions For The Authors:**

In general, I feel the authors make a good effort on making the paper easy to understand, however there are some questions I still have:

- While the problem is interesting, the experiments are somewhat limited. I understand that the main focus of the paper is to address the idea of evaluating knowledge-edited LLMs on multi-hop question, however the experiments only address three model (GPT-J, Vicuna-7B and GPT-3) which are all very different in size and how they were pre-trained. Is there a reason why a more diverse set of models (size, data, inductive biases, etc) were not used, as the results would be slightly more solid if these were also provided?

- Furthermore, Table 5 (and Table 7 in the appendix) appears to show comparisons of MEMIT and MeLLo only for GPT-J. I'm curious as to why Vicuna-7B and GPT-3 were included if performance with MEMIT (or other methods) are not provided as it is difficult to judge these rows. While I am already convinced by the performance of GPT-J, I think this part could be improved both for presentation purposes (elaborated upon further on) as well as for clarity.

- I feel there are some ablation studies that are missing, namely seeing how performance scales with the number of hops for MQuAKE. While I understand creating very large $k$-hop questions are difficult, I feel that having hops up to the teens would be much more interesting. I also would be interested in seeing more model sizes as a way of seeing how knowledge-editing methods scale with size as well.

**Reasons To Accept:**

- The problem this paper aims to address is very interesting and relevant, as current LLMs are very costly to train. As a result, knowledge editing is often considered a useful alternative to trying to keep models up to date.
- MeLLo is simple to understand and from my perspective a relatively effective way to ensure up-to-date retrieval of parametric knowledge from LLMs.

**Reasons To Reject:**

While I am pleased with the paper and see no major flaws, I'll list out a couple of minor issues I would only perhaps use if I had to be very nit-picky on the margins given a host of other papers I deem of similar quality.

- MeLLo requires storing information externally, therefore it would appear to me that it is only useful when a limited amount of facts need to be edited. In the event that perhaps a large portion of the model's knowledge needs to be edited, it would appear to me that MeLLo might not scale as well as other methods that are benchmarked against.
- MQuAKE is limited to only 2-4 hop questions, which would appear somewhat limited given the size of the models generally used nowadays.

**Reproducibility:**

4: Could mostly reproduce the results, but there may be some variation because of sample variance or minor variations in their interpretation of the protocol or method.

**Reviewer Confidence:**

4: Quite sure. I tried to check the important points carefully. It's unlikely, though conceivable, that I missed something that should affect my ratings.

**Typos Grammar Style And Presentation Improvements:**

Orthography and vocabulary seem to have no visible issues from my perspective.

In terms of presentation, my suggestions are quite minor. For example, I do believe that the swigly underlines in Table 1 could be instead changed with some form of highlighting (for example on the edited entity) since it could be more visible apparent what you're attempting to express.

---

> ### Author Rebuttal · Authors · 2023-08-29
>
> Thank you for your thoughtful discussion. We were encouraged that you see our new benchmark MQuAKE as an effective, easy-to-use way to evaluate model editing techniques and that you find our proposed method MeLLO to be simple and effective. Here we address your detailed comments, which are helping us to revise the paper and chart out future directions.
>
> **“MeLLo might not scale as well as other methods that are benchmarked against”**
>
> Scaling is certainly a vital issue in this area, and we see MeLLo as very competitive. The current state of the art is that most published model-editing methods degrade badly after over 100 edits, with only MEMIT (Meng et al. 2022) showing good performance out to 10,000 edits (and unfortunately they show no data beyond this point). While the number of “facts” in a language model is not clearly defined and unclear, it seems reasonable to assume that this is probably still less than 0.1% of the facts in even a 6B parameter LM.  Our experiments have shown that MeLLo is much more effective compared to all existing methods when editing thousands of facts at the same time (Table 5). Editing even more facts (e.g., 1M facts) is an open problem in this field, which we hope to explore in the future.
>
> **“MQuAKE is limited to only 2-4 hop questions”**
>
> We acknowledge this limitation, while emphasizing that the vast majority of the literature on multi-hop QA, such as work using the HotpotQA dataset, only considers 2 hops. We note that our goal is to assess whether edited models can answer questions where the answer should change as an entailment of edited facts. Our dataset consisting of 2-4 hop questions have shown the failures of existing baselines. The MQuAKE procedure extends relatively easily to additional hops, and it would be excellent to see the field progress to the point where 4 hops isn’t the serious challenge as it is now.
>
> **“Is there a reason why a more diverse set of models (size, data, inductive biases, etc) were not used?”**
>
> Due to limited compute resources and API budgets, we could not conduct experiments on all models. We would like to argue that we *did* choose a diverse set of base models, while nevertheless acknowledging that having results on even more models would be better.
>
> * We use GPT-J, a common LM that prior works (e.g., Meng et al., 2022) have evaluated on, as the base model in our main experiments.
>
> * We additionally consider a more recent and stronger model with a similar size, Vicuna-7B, in our experiments. Vicuna-7B is built based on the state-of-the-art (at submission time!) open-source language model LLaMA, and is representative of the many recent RLHF instruction-tuned models.
>
> * We further perform MeLLo on GPT-3 (text-davinci-003) to showcase that our approach is effective at editing large black-box language models, while previous approaches fail to do so.
>
> **“How performance scales with the number of hops for MQuAKE”**
>
> Thanks for the suggestion! We include breakdown results of evaluating ROME on MQuAKE-CF in the two tables below. We find that (1) the performance on 2-hop questions is much higher than 3-hop and 4-hop questions; (2) the performance gets worse when there are more edits associated with the test instance. We will include detailed breakdown results for other models in our revision.
>
> | 2-hop questions | 3-hop questions | 4-hop questions |  All |
> |:---------------:|:---------------:|:---------------:|:----:|
> |       33.8      |       9.1       |       11.4      | 18.1 |
>
> | Questions with 1 edited fact | Questions with 2 edited facts | Questions with 3 edited facts | Questions with 4 edited facts |  All |
> |:----------------------------:|:-----------------------------:|:-----------------------------:|:-----------------------------:|:----:|
> |             23.8             |              20.9             |              9.0              |              2.6              | 18.1 |
>
> **“On table 5, why Vicuna-7B and GPT-3 were included if performance with MEMIT (or other methods) are not provided as it is difficult to judge these rows”**
>
> Table 5 aims to show (1) MeLLo is more effective than all the baselines with the same base model, and (2) MeLLo performs better when using a stronger LLM (e.g., GPT-3). We include the performance of MEMIT for the purpose (1) and the full performance of MEMIT (and other models) is shown in Table 3 and Figure 2. We acknowledge that this may cause confusion for readers. We will present these results clearly in the revision.

---

### Meta-Review · Area_Chair_YMog · 2023-09-17

**Recommendation:** 5

**Metareview:**

All the reviewers rated the paper high on soundness as well as excitement. The paper was commended for well-thought motivation and being clearly written. All reviewers highlighted the potential impact of MQuaLE, a dataset of multihop questions collected by the authors to test the capabilities of knowledge editing techniques.

Reviewers acknowledged the significance of the findings reported in the paper. For instance, the experiments in the paper demonstrate that existing methods of knowledge editing might be able to recall individual edited facts but are unable to successfully use these updated facts to answer multi-hop questions using them. The paper also proposes MeLLo, a new memory-based approach to store all the facts externally, that outperforms past knowledge editing methods on MQuaLE.

There were a couple of concerns raised by the reviewers. For instance, the setup might be too simplistic and amenable to the retriever used in MeLLO. Another concern was the absence of prompt templates for the main experiments. Authors should consider addressing such questions in future versions of the paper.

---

### Decision · Program_Chairs · 2023-10-07

**Decision:**

Accept-Main

**Comment:**

All the reviewers rated the paper high on soundness as well as excitement. The paper was commended for well-thought motivation and being clearly written. All reviewers highlighted the potential impact of MQuaLE, a dataset of multihop questions collected by the authors to test the capabilities of knowledge editing techniques.

Reviewers acknowledged the significance of the findings reported in the paper. For instance, the experiments in the paper demonstrate that existing methods of knowledge editing might be able to recall individual edited facts but are unable to successfully use these updated facts to answer multi-hop questions using them. The paper also proposes MeLLo, a new memory-based approach to store all the facts externally, that outperforms past knowledge editing methods on MQuaLE.

There were a couple of concerns raised by the reviewers. For instance, the setup might be too simplistic and amenable to the retriever used in MeLLO. Another concern was the absence of prompt templates for the main experiments. Authors should consider addressing such questions in future versions of the paper.